# Fitness and the Crisis: Impacts of COVID-19 on Active Living and Life Satisfaction in Austria

**DOI:** 10.3390/ijerph182413073

**Published:** 2021-12-11

**Authors:** David Jungwirth, Chiara Amelie Weninger, Daniela Haluza

**Affiliations:** Department of Environmental Health, Center for Public Health, Medical University of Vienna, 1090 Vienna, Austria; davidjungwirth@gmx.at (D.J.); chiara.weninger@hotmail.com (C.A.W.)

**Keywords:** COVID-19 pandemic, physical activity, mental health, nature, sitting time

## Abstract

The COVID-19 pandemic has dramatically impacted human lifestyles across the world. Lockdowns and home confinement decreased prior opportunities for everyday physical activity. To retrospectively assess how the Austrian population coped with these aspects of the crisis, we conducted a cross-sectional online survey from March to September 2021 using a structured questionnaire in German. In total, 1214 participants (56.9% females, mean age 37.0 years) living across Austria shared self-reported information on sociodemographic characteristics, indoor and outdoor physical activity, reasons for being outdoors, and life satisfaction before and after the emergence of the virus. As a result, overall indoor physical activity significantly decreased in a before–during COVID-19 crisis comparison, although exercising at home with online instructions increased by about 63%. Exercising outdoors increased overall, specifically in periurban forests and rural areas, both by about 9%. Life satisfaction decreased significantly by 19.7% (*p* < 0.001). Outdoor public places and natural environments gained importance due to restrictions affecting access to sport facilities. Further research is needed to evaluate benefits and therapeutic values of outdoor nature for physical and mental health in times of a global pandemic to maintain resilient societies, as it might impact future active living and life satisfaction.

## 1. Introduction

Since March 2020, the outbreak of the newly detected severe acute respiratory syndrome coronavirus 2 (SARS-CoV-2), causing an infectious disease soon classified as Coronavirus Disease 2019 (COVID-19) has changed the world [1]. Rigorous attempts to control the spread of the virus and to protect national medical systems were imposed. These lockdowns and social distancing measures with massive restrictions on normal everyday social life brought nations to a sudden standstill—most obviously in work, education, sport, recreation, and travel [2].

In this article, we focus on changes in active living during the COVID-19 pandemic, anticipating the multifaceted impacts on (semi)professional sports, fitness, and recreational active living. Sport mega-events such as the 2020 Olympic Games in Tokyo, the 2020 UEFA European Football Championship, and countless tournaments in many other sports such as golf, tennis, Formula 1 motor racing etc. were postponed or, more often, canceled. In most countries, indoor and also outdoor sport facilities such as gyms, fitness centers, and sport clubs were closed for months, or permanently [3]. 

From a public health perspective, regular exercise is one of the most influential measures for health prevention. Physical activity improves overall quality of life, physical and mental health by positively affecting muscular and cardiorespiratory fitness, as well as bone and functional health, while reducing the risk of non-communicable disease, i.e., hypertension, coronary heart disease, stroke, metabolic syndrome, adiposity, various types of cancer, and depression [4,5]. Most recent international guidelines recommend a minimum required volume of moderate-to-vigorous intensity physical activity for health benefits of at least 150 min per week, with trends towards recommendations of 300 min per week [6]. Although the main focus of public health guidelines in view of the sedentary behavior towards increasing physical activity is the prevention of disease in the sense of primary prevention, regular exercising is also vital in rehabilitation. Currently, post-diagnostic physical activity is an important pillar of cancer treatment [7]. 

Systematic gender differences in frequency and motivation for physical activity were observed in different studies. Exemplarily, Molanorouzi et al. found that females seem to be motivated by physical condition and appearance, whereas men have more interest in challenges and competition, which could influence the reasons of exercising [8]. As a consequence, COVID-19-related changes to social life, canceled sports events, and closure of sports facilities could affect active living of females and males differently. To date, it is unknown on whether or how lifestyles of genders will be affected differently in a mid- to long-term range.

In times of urbanization and digitalization enforcing a sedentary lifestyle, people commonly spend too much time engaging in behaviors that consume low amounts of energy. Already in 2018, Guthold and colleagues described worldwide trends in insufficient physical activity, ultimately resulting in a range of chronic health conditions [9]. Recently, many international studies reported on the negative impact of the COVID-19 pandemic on quantity of physical activity, whereas screen-related sedentary behavior increased in both genders, and children and adults alike [10,11,12,13,14]. A study by Pieh and co-workers found a higher prevalence of mental health problems and lower quality of life and well-being in the first lockdown compared to rates before the COVID-19 pandemic in Austrian inhabitants, especially among females, younger adults, people without work, and those with lower income [12].

Addressing the inherent conflict of goals between home confinement and active living, the impact of the pandemic on physical activity gained high interest during the last months [15]. In the course of the crisis, evidence accumulated that life satisfaction significantly decreased during the pandemic, potentially due to decreased social activities with family and friends, restrictions in sports, and closed recreational facilities [16,17,18,19,20]. Given the immense negative consequences on psychosocial, mental, and physical health and well-being in all strata of the population, the closure in the sport, exercise, and leisure sectors, in addition to home schooling, home office, isolation, quarantine etc., created the need for innovative alternatives for being physically active. In their study on fitness enthusiasts, Kaur and co-workers found that although their motivation for exercising at home was low during the initial lockdown phase, participants were able to shift to home-based exercise quickly [21]. Thus, the online fitness industry has seen a major boom since the COVID-19 outbreak. Only within one year, this fitness branch was pushed from the no. 26 trend in 2020 to the top trend for 2021, as reported in the worldwide survey of fitness trends for 2021 [22]. 

The impacts of the COVID-19 pandemic were on a global scale, with many nations following similar general paths in fighting the spread of the pandemic. However, the constantly adapted lockdown measures mirrored the different pandemic waves and were not identical in intensity and duration, not even in neighboring countries or even federal states. This warrants a region-specific evaluation on the respective population-wise perceptions of the national regulations in force during that period. In Austria, measures against the spread of the virus became compulsory on 16 March 2020, i.e., the first lockdown in a series of lockdowns in response to infection waves [12]. Only the following five exceptions of the ban to enter public places applied: activities to avert an immediate danger to life, limb, or property; professional activity; shopping to cover necessary basic needs; care and assistance for people in need of support; and exercising outdoors alone and with pets or people living in the same household. Especially, the latter exception acknowledged prior studies showing that contact to outdoor nature can positively influence human health and wellbeing [16,23,24,25]. 

A rapidly increasing number of publications on societal and public health implications of the pandemic gradually add preliminary empirical data on the value of freely admissible urban and periurban public space, particularly green space, in taking up the role of commercial sport centers that are closed due to the pandemic [16]. As far as in accordance with the presently applicable official restrictions, public open spaces offer a legal, convenient, and attractive alternative for exercising alone, in small groups, or with personal trainers [15]. However, in reality, during the lockdowns, residents of Austrian larger cities such as the capital Vienna used inner-city parks and green spaces for recreational purposes, leading to sometimes overcrowded areas, contradicting social distancing rules [18,26].

Physical activity has been one of the most important public health issues in modern civilizations already before the pandemic, which is legitimately often labelled as a public health crisis. We can also speak of a fitness crisis for a large part of the population, with many questions emerging concerning the risk–benefit ratio of physical activity in crowded public areas and indoor facilities alike. By retrospectively assessing behavioral and perceptive changes from an Austrian perspective, this report adds to the current literature on health and well-being effects of the pandemic on the Austrian population novel evidence from 2021, the second year of the crisis [16,17,18]. In this cross-sectional study, we collected online data on use of indoor as well as outdoor sports facilities, reasons for being outdoors, and life satisfaction comparing the “before” with the “during” COVID-19 period. We surveyed a sample of Austrian residents to test whether the pandemic: (i) led to a shift of physical activity from indoor to outdoor spaces, (ii) changed reasons for being outdoors, (iii) impacted life satisfaction, and (iv) whether perceptions of females and males differed in these respects. 

## 2. Methods

### 2.1. Study Design

This non-representative, cross-sectional study aimed at identifying the self-reported prevailing perceptions regarding active living and life satisfaction in the context of the pandemic using an online questionnaire in German, targeting the German-speaking Austrian adult population. The online survey was pretested by 14 participants in February 2021 to review completeness and comprehensibility of the items. The adapted online survey was accessible barrier-free via the web-based survey tool SoSci Survey from March 8 until 8 September 2021 [27]. During the survey period, in Austria, the third lockdown was in place from 5 March to 19 May 2021, with relaxations to curfews and other restrictions lasting until August, albeit face masks were obligatory indoors and visitors of restaurants or events had to be fully vaccinated or recovered or (PCR or virus antigen) tested [12]. In August 2021, mandatory exit tests for high incidence areas were introduced in several Austrian districts. As incidence rates continuously increased to all-time highs and vaccination rates remained low, the fourth national lockdown for 20 days is currently in place and started on 22 November 2021. 

The cover page informed participants about the study aim, and their consent was implicitly obtained when completing the survey. Participants were informed that the period before the COVID-19 pandemic was defined as the time before 16 March 2020, when the first lockdown started in Austria [12]. A further explanatory text defined the intended meaning of indoor and outdoor sports facilities to ensure that participants referred to the correct type of facility: The term “indoor sports” means physical activity conducted within an enclosed building where people usually assemble to engage in physical exercise. The term “outdoor sports” refers to physical activity conducted in open space with both natural and artificial surfaces.

Using snowball sampling, the link was spread via commonly used social networks such as Facebook or WhatsApp of several public sports centers, organizations, clubs, and by private individuals such as national handball team members and trainers, and specifically as an online article by Sportunion Austria [28], and on the website of the Austrian Athletics organization [29]. We did not offer incentives for study participation. The institutional ethical committee of the Medical University Vienna, Austria, approved the study protocol on 4 March 2021, and the study was conducted following the ethical standards laid down in the Declaration of Helsinki.

### 2.2. Measures

The first part of the online survey collected data on socio-demographic characteristics age (in years), gender (male, female, divers), residence (Vienna, Lower Austria, other Austrian counties), and education level (primary, secondary, tertiary). We further assessed ratings on frequency of physical activity in six different indoor (fitness center, sports clubs, school/university sports center, at home independently, at home with (online) instructions) and outdoor (urban area, public green space, periurban forest, rural area) places. Each of these activities were separately assessed before and during the COVID-19 pandemic using a Likert-scale format: 1: never, 2: ≤1 h per week, 3: 1 h per week, 4: 1–2 h per week, 5: 3–5 h per week, and 6: ≥5 h per week. 

Moreover, participants rated six reasons for being outdoors (urge to move, recreation, social contacts, physical health, mental health, distraction) before and during the COVID-19 pandemic, using a 5-point Likert-scale format. The response options were: 1: strongly disagree, 2: disagree, 3: neutral, 4: agree, and 5: strongly agree. 

The last part of the questionnaire included the three items of the standardized Short Life Satisfaction Questionnaire for Lockdowns (SLSQL) to assess participants’ life satisfaction before and during the COVID-19 pandemic [19,20]. The respective items were “In most ways my life is close to my ideal.”, “So far, I have gotten the important things I want in life.”, and “I am satisfied with my life.”. The response options were: 1: strongly disagree, 2: disagree, 3: neutral, 4: agree, and 5: strongly agree. 

### 2.3. Statistical Data Analysis

We used descriptive statistics to report categorical data as absolute and relative frequencies, and continuous data as mean and standard deviation (SD). All statistical analyses were performed using the commercial statistical software SPSS Statistics for Windows, Version 27.0 (IBM Corp., Armonk, NY, USA). We employed paired samples T tests to calculate total as well as separately for females and males in scored responses between the before and during COVID-19 period. We summarized the three items of the SLSQL before and during the COVID-19 pandemic, respectively, to generate the total score Life Satisfaction for each of these two periods. We determined internal consistency for the three items of the SLSQL before and during the COVID-19 crisis, respectively, using Cronbach’s alpha. We further calculated the effect sizes to determine the magnitude of the change score using Cohen’s d and interpreted them as small (i.e., 0.2), moderate (i.e., 0.5), or large (i.e., 0.8), as well as lower and upper 95% confidence intervals (CI). Statistical significance was set at *p* < 0.05. 

## 3. Results

### 3.1. Study Population

The web link to the online survey was accessed 3161 times; 1486 participants started and 1243 of those fully completed the survey (83.7% completion rate). We excluded three data sets from people indicating to be of diverse gender, as we were interested in potential differences between males and females, and 26 data sets from participants living abroad, as we were interested in circumstances caused by the Austrian lockdowns, 12 from Germany, seven from Italy, three from Switzerland, and one person each from Luxembourg, the United Kingdom, Unites States of America, and Mongolia. Thus, the final sample included 1214 participants, with a higher proportion of females (*n* = 691, 56.9%) compared to males (*n* = 523, 43.1%, Table 1). Study subjects lived across all nine federal states of Austria, with 29.7% living in Vienna (*n* = 361), 32.5% in Lower Austria (*n* = 394), and the rest in the remaining parts of the country (*n* = 459, 37.8%). Further, 15.9% (*n* = 193) of participants reported having a primary, 41.2% (*n* = 500) a secondary, and 42.9% (*n* = 521) a tertiary education level. Average age of study participants was mean 36.92 years (SD 15.57, range 18 to 79 years). Average completion time of the survey was mean 6.0 min SD 2.2.

### 3.2. Changed Patterns of Physical Activity in the Pandemic

Table 2 reports participants’ self-reported frequency of physical activity at specific indoor (I) and outdoor (II) sports facilities before and during the COVID-19 pandemic, stratified by gender. Statistical analysis showed an overall statistically significant average decrease of 14.90% from mean 2.25 (SD 0.76) before to mean 1.92 (SD 0.65) during the COVID-19 pandemic for the frequency of indoor sports facility use (t = 18.377, *p* < 0.001), with a moderate effect size (Cohen’s d: 0.527, CI 95%: 0.467–0.587). In contrast, we found an overall statistically significant average increase of 5.29% from mean 2.28 (SD 0.90) before to mean 2.40 (SD 0.95) during COVID-19 for the frequency of outdoor sports facility use (t = –6.103, *p* < 0.001, d = −0.175, CI 95%: −0.232–−0.118).

Specifically, we revealed large decreases before vs. during the COVID-19 period and gender differences (all: *p* < 0.001) in the frequency of indoor sports activity at a fitness center (49.6%), a sport club (53.3%), and a school/university sports center (29.3%). In contrast, we found increased indoor sports activities at home independently as well as with online instructions (25.7 vs. 62.5%) and respective differences among the female and male groups (all: *p* < 0.001). Compared to males, females showed a strong increase for exercising at home with online instructions (42.75% vs. 73.81%). We found a small decrease before vs. during the COVID-19 pandemic in the frequency of outdoor sports activity in public green spaces (4.3%), which was only statistically significant in males (*p* = 0.018). We found a small increase in the frequency of outdoor sports activity in urban areas (4.8%), which was also only statistically significant in males (*p* = 0.007). Participants reported small increases of slightly over 9% in the frequency of outdoor sports activity in periurban forests and also rural areas, with statistically significant differences among the gender groups (all: *p* ≤ 0.002).

### 3.3. Reasons for Being Outdoors in the Pandemic

As shown in Table 3, we found a small but statistically significant increase of 5.51% for all reasons for being outdoors for the total sample from mean 3.91 (SD 0.68) before to mean 4.13 (SD 0.72) during the COVID-19 pandemic (t = −12.123, *p* < 0.001), with a small effect size (Cohen’s d: −0.348, CI 95%: −0.406–−0.290). The top three changes before vs. during the COVID-19 pandemic were identified for mental health (8.6%), distraction (7.8%), and urge to move (6.7%). Whereas all reasons statistically significantly increased in females, males only showed a statistically significant increase in the ratings for “recreation” and “social contacts”.

### 3.4. Life Satisfaction in the Pandemic

Table 4 summarizes the responses to the SLSQL, comparing ratings before and during the COVID-19 pandemic, stratified by gender. Internal consistency was moderate (Cronbach’s alpha 0.794) for the three items of the SLSQL before the COVID-19 pandemic and high (Cronbach’s alpha 0.846) for the three items of the SLSQL during the COVID-19 pandemic. Effect sizes ranged from moderate to large, with highest values for “In most ways my life is close to my ideal (I1)” (Cohen’s d: 0.844).

Statistical analysis showed that the total score of SLSQL, i.e., Life Satisfaction, decreased statistically significantly by 19.7% during compared to before the COVID-19 period (t = 27.5, *p* < 0.001, d = 0.79). We found highest changes in the total score of I1 with −25.8%, and observed statistically significant decreases in the total, and also in the female and the male samples for all items, with all *p* values from paired samples T tests smaller than 0.001. The total score Life Satisfaction was lower in males compared to females but with a higher change in males compared to females (−20.0 versus −19.4%).

## 4. Discussion

Although regular physical activity is essential for public health and mental well-being, many questions arose during the COVID-19 crisis concerning the risks of fitness-related activities in public areas and indoor sports facilities, mainly due to the perceived and actual infection risk. Guicciardi et al. proposed that fear of possible infection in indoor areas was the main reason for changed sports behavior in indoor facilities [30]. However, national customs along with governmental restrictions and lockdowns might impact sports behavior even more radically, arguing for an evaluation of national public perceptions not ultimately transferrable to other settings. Thus, this study examined how the current crisis impacts recreational activities indoors and outdoors, and life satisfaction of Austrian residents in 2021, the second year after the pandemic hit Austria in March 2020. Additionally, we assumed that modes of physical activity were differently affected in female and male participants.

Results of this study showed that physical activity decreased in Austria during the COVID-19 pandemic, as female and male participants spent less time on exercising. Similar results were found in earlier studies conducted in the beginning of the crisis [12,14,18]. Our results revealed that decreased exercising indoors mostly applied to public sport facilities (i.e., fitness centers or sports clubs), as home exercising gained popularity during the crisis. Thus, participants increasingly exercised at home independently or with online tutorials during the COVID-19 pandemic, suggesting that private homes were, all of a sudden, the most popular indoor category for exercising.

This observation is in agreement with the worldwide boom in using online fitness offers [21,22], and can be seen as a natural consequence of the already existing phenomenon of digital media and technology use for self-optimization in the Austrian population, also picturing a global trend [31,32,33]. Notably, our study found different pattern in female versus male physical activity during the COVID-19 crisis in Austria. Female participants reported to spend more time on exercising with online tutorials/classes, suggesting that the pronounced differences how females and males used home exercising for fitness developed during the COVID-19 pandemic or were accelerated by it. These observations might be connected to different motivational factors for sports, with females being more likely to use physical activity for motives associated with enhanced wellness, physical appearance, and attractiveness, and males being more interested in muscle mass-building exercises, fitness equipment, and team competition games [8].

On a global level, the COVID-19 home confinement had a negative effect on all physical activity intensity levels from vigorous to low, while daily sitting time increased from five to eight hours per day [19]. Kim et al. reported that the frequency of aerobic exercise was lower, while the frequency of anaerobic exercise was higher during the COVID-19 pandemic in Korea [34]. This observation is presumably transferable to our own results on the increased use of online tutorials that potentially integrate aspects of anaerobic high-intensity functional or high-intensity interval training to time effectiveness and resource-saving improved fitness [35].

According to our study, outdoor sports gained popularity among both male and female participants during the COVID-19 pandemic. As for outdoor sports, we found a less pronounced increase for visiting urban areas, periurban forests as well as rural areas by about 9%, as participants seized the opportunity to use natural environments for physical activity. Participants mentioned to use being outdoors intentionally for various health and wellbeing aspects before the COVID-19 crisis. During the pandemic, the majority of the participants considered health and physical activity as important motivational factors for visiting outdoor areas. These findings are not surprising, as prior studies suggested increased importance of nature for regular physical activity during the COVID-19 pandemic [18,26,36]. Consequently, the value of recreational outdoor activities has presumable increased during the crisis. These findings are supported by Park et al., revealing that the bicycle use in Seoul, South Korea, doubled during the COVID-19 pandemic [36].

Prior studies highlighted the recreational value of forests and urban parks before and also during the crisis [23,24,25,26]. Robinson et al. found that people spent more time in nature for health and wellbeing benefits and felt that nature helped them to manage the COVID-19 pandemic [37]. In urban populations, access to outdoor spaces and nature views increased mental health during lockdowns [38]. Our results suggest that study subjects used urban parks less for sports and exercising, but nature and hiking areas were more popular during the pandemic compared to before.

We were interested in how the crisis changed perceived life satisfaction among our study sample, as measured by the Short Life Satisfaction Questionnaire—Lockdowns [19,20]. The total score of SLSQL, i.e., Life Satisfaction, decreased by 19.7% during compared to before the COVID-19 pandemic and was lower in males compared to females, but with a higher change in males compared to females (−20.0 versus −19.4%). These figures correspond to a global trend in increased insecurity and dramatic restrictions of social life and are similar to other reports, e.g., from Ammar et al., who showed that life satisfaction decreased by 16% during the COVID-19 pandemic in a global evaluation [19,20].

It remains unclear yet whether life satisfaction scores will notably stabilize or increase to pre-COVID-19 crisis levels in the nearest-to-near future. Mental health problems, triggered by home confinement during the COVID-19 pandemic globally, correspond to increased fear, insomnia, depressive symptoms, anxiety, and stress [39]. To mitigate the risk of long-lasting stigma and isolation, barrier-free therapeutic support for people with vulnerable mental health status during and after isolation and lockdowns are inevitable.

Notably, changes in life satisfaction scores were similar in females and males, although previous studies suggested that women were more affected by anxiety and depression, as exemplarily shown by a study conducted in the first wave of the COVID-19 pandemic in Spain [40]. Similar to previous research pointing out that social distancing decreased life satisfaction [20], findings of our study suggest a negative impact of the COVID-19 pandemic on life satisfaction in our Austrian sample. In this respect, Ammar et al. highlighted the importance of outdoor environments for social contacts, as participants reported to visit outdoor areas more often for social activities [20]. Likewise, Robinson et al. considered nature experience as a beneficial for health and wellbeing in the pandemic [37].

The current virus outbreak has forced us to think about the risks of future pandemics or other global crises, and how we may increase our resilience to withstand their short-, medium- and longer-term impacts in all areas of social life. Lockdowns, social distancing, self-isolation, quarantine, home schooling, teleshopping, and teleworking during a pandemic contradict an active lifestyle, and are likely to contribute to physical inactivity [10,12,14,18,19,21,30,36,41], and subsequently in lower life satisfaction [19,20]. Combining the need for novel routines in active living in view of closed fitness facilities with a new awareness for the value of urban and rural green areas, life satisfaction could be eventually restored to a pre-pandemic level. These thoughts are supported by Folk et al., who emphasized the need of new health concepts and reviewed fitness guidelines, anticipating that future public health measures will be affected by the aftermaths of the COVID-19 crisis [42].

A clear understanding of how people can be empowered and motivated to self-determinedly break the vicious circle of longer-term and repeated home confinement and inactivity is needed. In a following step, targeted public health interventions should be designed to mitigate the negative effects of the COVID-19 pandemic or future pandemics on physical activity and life satisfaction [41]. Along with wider-reaching national strategies and imbursement plans for systemically relevant professional groups, educational institutions should offer sport programs for educators and students, and companies should be aware of the necessity to address the specific needs of their employees to maintain high physical and mental performance and productivity in challenging times [42].

## 5. Limitations

Our study had limitations that should be kept in mind when interpreting its results. As this study was designed as a non-representative, cross-sectional online survey, Internet access was necessary to participate in the survey, leading to a selection bias. Sports organizations were asked to share the link to the survey to the reach into all different federal states, age, and gender groups. With the additional recruitment through snowball sampling, we assured to reach a broad strata of the population, as Eysenbach and Wyatt claimed this to be the most essential factor in minimizing selection bias [43]. The anonymous nature of the online survey did not allow for investigating potential reasons for non-response. We relied on participants’ self-reports, which introduced recall bias and can potentially lead to an overestimation of physical activity. The detailed types and intensity of physical exercise were beyond the scope of this assessment, mainly focusing on the shift of modes of activity from indoor to outdoor environments. However, there is abundant evidence that all levels of physical activity ranging from low to high decreased due to COVID-19 home confinement, while sedentariness increased [10,12,14,18,19,21,30,36,41].

We collected data over a period of time (from March until September 2021) during an unprecedented time during a global pandemic. As the epidemiological situation changed, and still changes, dynamically and unpredictably, the answers given by the respondents in April might differ from those in July. However, this retrospective study focused on self-reported data comparing a time period before the first lockdown measures in March 2020 to a time period after this event. There is the possibility of unmeasured confounding associated with opinions on the COVID-19 pandemic such as lack of interest or even resistance to acknowledge the crisis. Most participants were highly educated and rather young, with an average age of about 37 years, limiting generalizability of the results and representativeness of our study population to the entire population. To further elucidate reasons for differences in the perceptions of gender in active living and life satisfaction of the Austrian population during the COVID-19 pandemic, further research using mixed-methods studies and longitudinal study designs should be conducted.

## 6. Conclusions

Regular physical activity is an important public health issue, with its positive effects on physical and mental health being recognized since ancient times. Despite the importance of sports, our non-representative cross-sectional online study revealed decreased physical activity and life satisfaction during the COVID-19 pandemic in a quite large sample of Austrian inhabitants. Further, outdoor experience gained popularity, although to a lesser extent than anticipated, whereas home-based online classes boomed due to the closure of indoor sports facilities and other venues of social life.

As overall active living decreased during the crisis, public health messages should highlight the value of outdoor sports and nature contact for physical and mental health. Further, sport programs in outdoor environments should be supported to enlarge the current options for regular physical activity and prevent eventual post-pandemic health disparities. These multisector measurements require the cooperation among different stakeholders (i.e., health experts, sports organizations, government, civil society, city planners) and substantial financial support for hygiene concepts to support sports clubs and reduce fear and actual as well as perceived infection risk.

While recognizing the importance of restricting mobility to contain the COVID-19 pandemic, a bundle of nation-wide behavioral strategies for preventing global acute and chronic physical and mental health effects during lockdowns is key. We strongly recommend designing, implementing, and evaluating national programs for safe outdoor or home-based approaches for regularly and repetitively disrupting physical inactivity in all strata of the population.

## Figures and Tables

**Table 1 ijerph-18-13073-t001:** Sociodemographic characteristics of the study population (*n* = 1214).

	*n*	%
I. Gender		
Female	691	56.9
Male	523	43.1
II. Education level		
Primary education	193	15.9
Secondary education	500	41.2
Tertiary education	521	42.9
III. Geographical region		
Lower Austria	394	32.5
Vienna	361	29.7
Styria	98	8.1
Upper Austria	95	7.8
Vorarlberg	86	7.1
Salzburg	55	4.5
Tyrol	47	3.9
Burgenland	40	3.3
Carinthia	38	3.1

**Table 2 ijerph-18-13073-t002:** Frequency of physical activity at specific indoor (I) and outdoor (II) sports facilities before vs. during the COVID-19 pandemic, stratified by gender.

Items	Total (*n* = 1214)					Females (*n* = 691)					Males (*n* = 523)				
	Before COVID-19		During COVID-19			Before COVID-19		During COVID-19			Before COVID-19		During COVID-19		
	Mean	SD	Mean	SD	Diff. %	Mean	SD	Mean	SD	*p*	Mean	SD	Mean	SD	*p*
I. Indoor sports facilities															
Fitness center	2.48	1.85	1.25	0.94	−49.60	2.43	1.79	1.22	0.87	<0.001 **	2.54	1.92	1.3	1.02	<0.001 **
Sports clubs	3.04	1.94	1.42	1.18	−53.29	2.99	1.89	1.41	1.16	<0.001 **	3.12	2.01	1.44	1.2	<0.001 **
School/ university sports center	1.57	1.28	1.11	0.54	−29.30	1.65	1.37	1.13	0.6	<0.001 **	1.46	1.14	1.08	0.45	<0.001 **
At home independently	2.65	1.58	3.33	1.71	25.66	2.6	1.51	3.22	1.67	<0.001 **	2.71	1.66	3.48	1.76	<0.001 **
At home with online instructions	1.52	1.09	2.47	1.66	62.5	1.68	1.17	2.92	1.66	<0.001 **	1.31	0.94	1.87	1.46	<0.001 **
II. Outdoor sports facilities															
Urban area	1.66	1.27	1.74	1.34	4.82	1.56	1.16	1.67	1.25	0.007 *	1.8	1.39	1.84	1.44	0.320
Public green space	2.1	1.54	2.01	1.51	−4.29	1.95	1.41	1.91	1.43	0.365	2.29	1.69	2.14	1.61	0.018 *
Periurban forest	2.05	1.5	2.24	1.66	9.27	1.99	1.43	2.23	1.63	<0.001 **	2.12	1.59	2.25	1.71	0.002 *
Rural area	3.32	1.76	3.62	1.82	9.04	3.29	1.71	3.68	1.77	<0.001 **	3.36	1.83	3.53	1.88	<0.001 **

Note: Means and SD; frequency of physical activity: 1: never, 2: ≤1 h per week, 3: 1 h per week, 4: 1–2 h per week, 5: 3–5 h per week, and 6: ≥5 h per week; *p* values from paired samples t test, * *p* < 0.05, ** *p* < 0.001.

**Table 3 ijerph-18-13073-t003:** Reasons for being outdoors before and during the COVID-19 pandemic, stratified by gender.

Items	Total (*n* = 1214)					Female (*n* = 691)					Male (*n* = 523)				
	Before COVID-19		During COVID-19			Before COVID-19		During COVID-19			Before COVID-19		During COVID-19		
	Mean	SD	Mean	SD	Diff. %	Mean	SD	Mean	SD	*p*	Mean	SD	Mean	SD	*p*
Urge to move	3.98	1.15	4.25	1.10	6.72	3.90	1.15	4.28	1.06	<0.001 **	4.09	1.14	4.22	1.15	0.002 *
Recreation	4.27	0.83	4.36	0.90	2.02	4.35	0.77	4.45	0.84	0.020 *	4.17	0.88	4.24	0.95	0.072
Social contacts	3.52	1.14	3.61	1.34	2.64	3.66	1.10	3.87	1.28	<0.001 **	3.34	1.18	3.27	1.35	0.251
Physical health	3.82	1.12	4.02	1.09	5.24	3.83	1.11	4.09	1.04	<0.001 **	3.81	1.14	3.93	1.14	0.002 *
Mental health	4.02	1.05	4.36	0.93	8.57	4.06	1.05	4.49	0.83	<0.001 **	3.96	1.05	4.19	1.03	<0.001 **
Distraction	3.86	1.07	4.17	1.09	7.83	3.96	1.03	4.32	1.01	<0.001 **	3.73	1.12	3.97	1.17	<0.001 **

Note: Means and SD; agreement: 1: strongly disagree, 2: disagree, 3: neutral, 4: agree, 5: strongly agree; *p* values from paired samples t test, * *p* < 0.05, ** *p* < 0.001.

**Table 4 ijerph-18-13073-t004:** Responses to the Short Life Satisfaction Questionnaire—Lockdowns (SLSQL) before and during the COVID-19 pandemic, stratified by gender.

Items	Before COVID-19		During COVID-19		Diff.	−Diff. %	T Test *	Cohen’s d	Lower 95% CI	Upper 95% CI
	Mean	SD	Mean	SD						
I1. In most ways my life is close to my ideal.										
Total	3.93	0.78	2.92	1.09	1.02	25.84	29.422	0.844	0.779	0.910
Females	3.93	0.80	2.95	1.09	0.98	24.99	21.699	0.825	0.739	0.912
Males	3.94	0.75	2.88	1.09	1.06	26.95	19.878	0.869	0.768	0.970
I2. So far, I have gotten the important things I want in life.										
Total	3.83	0.92	3.28	1.14	0.54	14.19	18.219	0.523	0.463	0.583
Females	3.86	0.91	3.32	1.13	0.54	13.93	13.837	0.526	0.447	0.606
Males	3.79	0.93	3.24	1.15	0.55	14.55	11.851	0.518	0.427	0.609
I3. I am satisfied with my life.										
Total	4.21	0.77	3.42	1.17	0.80	18.89	23.834	0.684	0.621	0.746
Females	4.22	0.78	3.41	1.16	0.81	19.19	18.135	0.690	0.607	0.773
Males	4.21	0.76	3.43	1.19	0.78	18.50	15.454	0.676	0.580	0.770
I1–I3. Total score Life Satisfaction										
Total	3.99	0.69	3.21	0.99	0.79	19.67	27.548	0.791	0.726	0.855
Females	4.00	0.71	3.22	0.98	0.78	19.40	20.91	0.796	0.710	0.881
Males	3.98	0.68	3.18	1.00	0.80	20.04	17.931	0.784	0.686	0.882

Note: CI: confidence interval; agreement with the items: 1: strongly disagree, 2: disagree, 3: neutral, 4: agree, 5: strongly agree; * all *p* values from paired samples T tests: < 0.001.

## Data Availability

The data supporting the findings of this study are available from the corresponding author upon request.

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
