# Peer review of "Fitness and the Crisis: Impacts of COVID-19 on Active Living and Life Satisfaction in Austria"

_ijerph, 2021, doi:10.3390/ijerph182413073_

Round 1

Reviewer 1 Report

The theme of the work carried out is pertinent and falls within the scope of the International Journal of Environmental Research and Public Health.

The study is very interesting and explores a relevant issue in society. Studying and analyzing the impacts of COVID-19 on active living and life satisfaction is an accurate approach.

Study design. The data was collected over a long period of time (from March until September), during which the epidemiological situation was changing dynamically. It is worth noting that the answers given by the respondents in April 2020 may differ significantly from those collected in July.

Results. The results section should be divided into sub-parts. In the sub-section describing study population it would be suggested to design a table with the sociodemographic data and to be congruent with the style of the results section. Sub-sections devoted to the subsequent analysis should be distinguished. In the descriptions of tables included in the text, an erroneous reference to Figures appears. Perhaps it is worth including the description of the results under (not over) the tables.

I have the impression that the wording “decreased statistically significantly by -19.7%” is incorrect (tautology).

Some ideas that have been signaled in the discussion can be developed, eg.

 “Female participants reported to spend more time on exercising with online tutorials/classes, suggesting that the pronounced gender differences in home exercising developed during the COVID-19 pandemic - or were accelerated by it. These observations might be connected to different motivation factors for sports and restricted indoor sports offers during lockdowns and home confinement.”

Reviewer 2 Report

I think this is a relevant and well written manuscript. There are mostly some small issues, that should be addressed and one bigger issue regarding the analyses. 

l. 15: I think you cannot speak from a pre-POST comparison, since the pandemic is not over yet. In general, you should speak of Covid-19 PANDEMIC or Covid-19 LOCKDOWN or something like that. You do not investigate the effect of the illness.

l. 27: COVID 19 is the illness, SARS-CoV-2 is the virus

ll. 49-53. The gender-aspect do not really fit in this paragraph. As gender is a major aspect in the analysis, the given information to gender-specific differences regarding physical activity should be expanded in an extra paragraph.

l. 76. Who is “we”?

l.114: It should be more explicit, what measure to contain the pandemic were in force during the survey-period.  

ll. 167 ff. I think you should conduct paired samples ANOVAs with gender as between subject factor for identifying an interaction effect of “time” and gender and then use post-hoc tests with correction for type I error. This should be done with all analyses including between-subject factors. In addition, this type of analysis allows differences to "t1" (before pandemic) to be taken into account and analyzed. The increases and decreases in certain variables should be interpreted against the background of potentially different initial values. 

195.ff. I do not know the reporting standards for this journal, but I think there should be a “=” after the SDs and a “;” between the CIs.

Table 1. I think the p-values which are reported as 0.001 are truly < 0.001.

ll.228 ff. Please provide a measure for the internal consistency of the SLSQL (e.g. McDonalds Omega)

279ff. Gamification as well as a young population were not investigated. Therefore, this interpretation goes too far.
